# Oligomer Formation by Amyloid-β42 in a Membrane-Mimicking Environment in Alzheimer’s Disease

**DOI:** 10.3390/molecules27248804

**Published:** 2022-12-12

**Authors:** Terrone L. Rosenberry, Huan-Xiang Zhou, Scott M. Stagg, Anant K. Paravastu

**Affiliations:** 1The Departments of Neuroscience and Pharmacology, Mayo Clinic, Jacksonville, FL 32224, USA; 2Departments of Chemistry and Physics, University of Illinois Chicago, Chicago, IL 60608, USA; 3Institute of Molecular Biophysics, Florida State University, Tallahassee, FL 32306, USA; 4Department of Biological Sciences, Florida State University, Tallahassee, FL 32306, USA; 5School of Chemical and Biomolecular Engineering, Georgia Institute of Technology, 311 Ferst Drive NW, Atlanta, GA 30332, USA

**Keywords:** Alzheimer’s disease, Aβ42, oligomer, solid-state 2D NMR, membranes

## Abstract

The brains of Alzheimer’s disease (AD) patients contain numerous amyloid plaques that are diagnostic of the disease. The plaques are primarily composed of the amyloidogenic peptides proteins Aβ40 and Aβ42, which are derived by the processing of the amyloid pre-cursor protein (APP) by two proteases called β-secretase and γ-secretase. Aβ42 differs from Aβ40 in having two additional hydrophobic amino acids, ILE and ALA, at the C-terminus. A small percentage of AD is autosomal dominant (ADAD) and linked either to the genes for the presenilins, which are part of γ-secretase, or APP. Because ADAD shares most pathogenic features with widespread late-onset AD, Aβ peptides have become the focus of AD research. Fibrils formed by the aggregation of these peptides are the major component of plaques and were initially targeted in AD therapy. However, the fact that the abundance of plaques does not correlate well with cognitive decline in AD patients has led investigators to examine smaller Aβ aggregates called oligomers. The low levels and heterogeneity of Aβ oligomers have made the determination of their structures difficult, but recent structure determinations of oligomers either formed or initiated in detergents have been achieved. We report here on the structures of these oligomers and suggest how they may be involved in AD.

## 1. Introduction

The involvement of amyloid in Alzheimer’s disease (AD) was first observed by neuropathological analysis (see [1]). The proteins comprising the two brain lesions diagnostic of AD are the senile (amyloid) plaques and the neurofibrillary tangles. The major constituent of the extracellular amyloid plaques, which occur in large numbers in brain areas important for memory and cognition, is the 40–42-residue amyloid β-protein (Aβ), derived from the β-amyloid precursor protein (APP). The intraneuronal neurofibrillary tangles in the same brain regions are composed of the microtubule-associated phosphoprotein tau [2]. APP is a single transmembrane pass protein that undergoes catabolic processing by proteases called secretases. The first step in the amyloidogenic pathway involves β-secretase, which cleaves APP predominantly at the N-terminus of Aβ to release sAPPβ, a ~100-kD soluble N-terminal fragment, and C99, a 99-residue C-terminal fragment that remains membrane bound [3]. A second step in this pathway involves an intramembranous, multisubunit protease called γ-secretase. This protease initially cleaves C99 to generate N-terminal 49- or 48-residue peptides and then, in processive cleavages from the C-terminus of these peptides, yields Aβ40 and Aβ42 [4,5]. These Aβ40 and Aβ42 peptides are amyloidogenic because they aggregate to form amyloid fibrils in both the brain and in vitro.

A small percentage of AD (~1%) is genetically transmitted within families and results in early-onset AD. This form of the disease is autosomal dominant (ADAD) and linked to one of only three genes: presenilin-1 or presenilin-2, which are alternative components of the γ-secretase complex, or APP [6]. Although ADAD constitutes only a small fraction of all AD cases, it is a critically important area of study because the pathological features of the disease are like the more common late-onset form. Down syndrome is associated with triplication of chromosome 21, and the gene for APP also resides on this chromosome. Chromosome 21 triplication results in Aβ overproduction and increased amyloid accumulation [7].

APP mutations increase Aβ production by different mechanisms. One double mutation (the Swedish mutation) immediately upstream of the β-secretase cleavage site amplifies cleavage by β-secretase, generating increased Aβ40 and Aβ42 from APP [8]. APP mutations at APP residues 714–717 (C99 residues 43–46) around the γ-secretase cleavage sites result in modification of γ-secretase activity, enhancing the production of Aβ42 relative to Aβ40 [9]. Mutations within the APP sequence (C99 residues 21–23) tend to increase the propensity of Aβ to aggregate [10]. Presenilin-1 and -2 mutations alter the conformation of the γ-secretase complex, increasing production of Aβ42 from APP [11].

## 2. A Diverse Group of Amyloid-β Aggregates

Early biological evidence for the hypothesis that amyloid aggregates are involved in the etiology of AD was observed (1) in the accelerated rate of disease progression in ADAD and Down syndrome patients and (2) the progressive accumulation of Aβ plaques in both ADAD and late-onset AD [12,13]. The plaques were detected by neuropathological analysis at autopsy, and the primary component of the plaques were the fibril forms of Aβ40 and Aβ42. Strong additional support for this hypothesis was derived from subsequent identification in ADAD of mutations in APP and presenilin-1 and -2. The most frequently voiced objection to the amyloid hypothesis is that the number of amyloid deposits in the brain does not correlate well with the degree of cognitive impairment that the patient experienced in life [13]. A response to this objection notes that non-fibrillar amyloid aggregates may be more toxic than fibrils. These aggregates include oligomers consisting of fewer than ~50 Aβ molecules and protofibrils including hundreds of molecules. Oligomers and protofibrils are too small to be readily detected by pathological analysis and appear to have widespread toxicity greater than that of monomers or fibrils. Endogenous Aβ oligomers have been detected, albeit at low levels, and exogenous Aβ oligomers have been prepared in numerous ways and show various neurotoxic effects [14]. More recently, widespread use of biomarker immunoassays for Aβ and PET scans for fibrils in living individuals permitted observation of the time course of Aβ production [15,16]: The earliest detection of elevated Aβ in the cerebrospinal fluid can precede the death of patients from AD by decades. There are many unknowns regarding what Aβ aggregates could be formed over the course of many years and how these aggregates could affect the brain.

Knowledge of Aβ aggregate structures is discussed with fibrillar structure as a basis since fibrils are the best-studied aggregates. Fibrils are filamentous structures with typical widths of ≈10 nm and lengths up to several micrometers. The first molecular structural model for an amyloid fibril to be based on significant high-resolution structural constraints was obtained for Aβ40 based on a set of experimental constraints from solid-state NMR spectroscopy [17]. The cross-β unit is a double-layered structure, with in-register parallel β-sheets formed by residues 12–24 and 30–40 (Figure 1). Amyloid fibril structures, however, are polymorphic. Differences seen between Aβ40 fibrils [18] and between Aβ40 and Aβ42 fibrils [19] can mostly be described in terms of variation in the residues involved in β-strand regions and the stacking of different β-sheets. Although most data on fibrillar structures report in-register parallel β-sheets, the Iowa mutant of Aβ40 is also capable of forming fibrils composed of antiparallel β-sheets. Based on FTIR, a low-intensity peak at 1685–1695 cm^−1^ has been proposed to be indicative of an antiparallel structure; this feature is present in Aβ oligomers but not in Aβ fibrils [20]. Neither endogenous oligomers nor endogenous protofibrils have been isolated with sufficient abundance or structural homogeneity for structure determination. Exogenous oligomers formed in detergents (noted below) have been characterized, and it appears that patterns of β-strand alignment, including coexistence of parallel and antiparallel β-sheets, may distinguish non-fibrillar from fibrillar structures. It also appears that some oligomers assemble along pathways that are incompatible with β-sheet elongation to fibrillar dimensions. Protofibrils, in contrast, may be on-pathway to fibril formation.

Present knowledge of oligomer structure is derived from experimental protocols that promote nonfibrillar aggregation pathways. In vivo, Aβ aggregation is stimulated at cellular interfaces rich in lipid rafts [21,22], and in vitro detergent micelles that promote Aβ aggregation may be good models of cellular interfaces [23]. Aβ aggregation is very sensitive to the concentration of the detergent sodium dodecyl sulfate (SDS) [24,25], and only Aβ42 and not Aβ40 formed a β-structured aggregate in SDS that failed to show fluorescence with thioflavin T (an indicator of fibril formation) [24,25]. Size exclusion chromatography (SEC) of this Aβ42 preparation revealed two peaks (Figure 2A) [26,27]. SDS-polyacrylamide gel electrophoresis (SDS-PAGE) (Figure 2B) revealed a constant ratio of four bands across the first peak, and monomer alone in the second peak. This result indicates that the dialyzed oligomers elute as a single species on SEC but that this species is partially dissociated by the SDS-PAGE loading buffer. The oligomer peak was characterized by multiangle light scattering (MALS) to give a nearly constant molecular weight *M* of 150-kDa [26].

## 3. Analysis of 150-kDa Oligomers by Solid-State 2D NMR

To confirm the apparent homogeneity of the 150-kDa Aβ42 oligomer detected by SEC and MALS, we examined lyophilized samples by solid-state NMR [28,29,30]. Aβ42 fibril and 150-kDa oligomer samples were uniformly labeled with ^13^C at 4 to 7 selected residues, and secondary structure was analyzed through backbone ^13^C chemical shifts and inter-residue proximities. Correlation of backbone ^13^C chemical shifts with those of known NMR-derived structures revealed two β-strands in the oligomers: an N-terminal strand (or N-strand) spanning residues 10–24 and a C-strand spanning residues 30–42 [30]. The most prominent structural differences between Aβ42 oligomers and fibrils were observed through measurements of intermolecular ^13^C-^13^C dipolar couplings observed in PITHIRDS-CT [31] experiments. PITHIRDS-CT data indicate that, unlike fibrils, oligomers are not characterized by in-register parallel β-sheets [28]. We also employed the 2D dipolar-assisted rotational resonance (DARR) technique [29,32] that produces off-diagonal peaks (cross peaks) corresponding to ^13^C atoms that are close (<6 Å) to one another. Sets of inter-residue cross peaks indicating contacts between M35 and G37 and between I32 and V40 were obtained, supporting a model where the C-strands form an antiparallel β-sheet centered at V36 [29]. However, neither PITHIRDS-CT nor 2D DARR spectra supported antiparallel or in-register parallel β-sheets in the N-strands. Instead, the 2D NMR data supported a parallel N-strand β-sheet shifted three residues out of register [30] (see Figure 3).

Constructing a model of the molecular structure from the 2D NMR data in Figure 3 was very challenging, and we turned to cryo-electron microscopy (cryo-EM) to obtain additional insight. A sample of 150-kDa oligomers was embedded in ice, and a pore was visible in many of the particles. We picked the particles and produced class averages using cryoSPARC [33]. The analysis revealed a class average with four-fold symmetry and a pore in the center [34]. Although the class averages had few features (likely due to the small size and the particles’ tendency to associate with thick ice), the four-fold symmetry together with the 2D NMR data allowed the construction of a model. All-atom models using molecular dynamics simulations allowed us to introduce experimentally inspired artificial restraints on backbone torsion angles and inter-atomic proximities [35,36]. Replicate modeling trials helped us understand what conformations are possible in regions of the structure that are not constrained by our experimental data. The modeling of the 150-kDa oligomer structure optimized a 32-mer (octamer of tetramers) with domain-shifted parallel N-strands in half of the subunits [34]. To summarize these and other results, more than fifteen samples of the 150-kDa oligomer were each labeled with ^13^C at 4 to 7 amino acids chosen to probe specific structural features. Figure 4 compares 2D DARR contacts predicted for our model (see [30,34]) (color scale) with tested inter-residue contacts that were observed (stars) or not observed (circles). The pattern of stars clusters near the diagonal because our preliminary experiments focused on the arrangements of β-strands. The N-strand contacts were consistent with a registry shift of ±3 [30]. The C-strand contacts were mostly on a line perpendicular to the diagonal and consistent with antiparallel β-strands centered on V36. Work currently in progress will address discrepancies between predicted and observed 2D DARR contacts.

## 4. A Tetrameric Aβ42 Oligomer May Be an Intermediate to 150-kDa Oligomer Formation

A caveat to our analysis of 150-kDa Aβ42 oligomers is that the structure was promoted by incubation in dilute SDS, which is typically known as a denaturant. We were therefore extremely interested in a report by Natalia Carulla’s group on the structure of an Aβ42 oligomer generated in a zwitterionic detergent, dodecylphosphocholine (DPC) [37]. Like our 150-kDa oligomer, their oligomer was only formed by Aβ42 and not by Aβ40. Employing solution NMR, they conducted 2D [^1^H–^15^N] TROSY measurements to deduce two types of Aβ42 peptides within the oligomer. They concluded that DPC induced an Aβ42 tetramer comprised of a β-sheet core made of six antiparallel β-strands, connected by only two β-turns, leaving two short and two long, flexible N-termini (Figure 5). 3D NH-H, NH-CH_3_, and CH_3_-CH_3_ NOESY contacts produced over 150 intramolecular pairings (NOEs) which were obtained to verify the structure. Remarkably, the two central β-strands contained residues 30–41 arranged in antiparallel fashion and centered on residue 36, identical to the C-strand arrangement in our model of the 150-kDa oligomer. The β-strands in the tetramer adjacent to the central β-strands were comprised of residues 30–41 arranged antiparallel to the central β-strands and shifted by one residue in register. All residues on both faces of the β-sheet core were hydrophobic except for three basic residues on the edge β-strands (H13, H14, and K16) and interacted closely with the detergent molecules.

The Carulla group did not report on detergent removal, but we speculate that their tetramer is a precursor to our 150-kDa oligomer and that our oligomer will be formed on DPC removal by dialysis. The formation of identical oligomer structures following removal of micelles composed of anionic (SDS) or zwitterionic (DPC) detergents would suggest that their important feature is the micelle hydrophobicity. This raises the larger question of the possible involvement of Aβ42 with phospholipid membranes.

## 5. Aβ42 Interaction with Phospholipid Membranes

Incorporation of Aβ into lipid membranes has been investigated for over 30 years [38,39]. Synthetic Aβ has been introduced to liposomes or planar phospholipid bilayers, and bound Aβ has been measured. Cryo-electron tomography was recently employed by Tian et al. [40] to reveal Aβ oligomers and curvilinear protofibrils binding extensively to synthetic lipid vesicles, inserting and carpeting the outer leaflet of the bilayer. Aβ monomers and fibrils interacted with phospholipid bilayers to a much lesser extent, if at all. The nature and extent of the interactions with Aβ oligomers and curvilinear protofibrils depend on the Aβ oligomer preparation method and the phospholipid composition. Tian et al. [40] examined a continuous Aβ incubation, taking monomers at early initial times, fibrils at late final times, and oligomers and curvilinear protofibrils at intermediate times. In an earlier study, this group also reported on pore formation by the same three preparations of Aβ in cell membranes [41]. Patch clamp membranes were excised from HEK293 cells of neuronal origin, and each Aβ preparation was free to diffuse toward the extracellular face of the membrane within the pipette. Only the Aβ42 oligomer/curvilinear protofibril preparation formed voltage-independent, non-selective ion channels. Aβ42 monomers, fibrils, and the Aβ40 oligomer/curvilinear protofibrils yielded no detectable channels. For the Aβ42 oligomer preparation, three instances of channel conductance were observed, corresponding to pore sizes of 1.7, 2.1, and 2.4 nm. We estimate a similar pore size for our 150-kDa oligomers [34].

While it is possible that our 150-kDa oligomers could interact with phospholipid membranes in a fashion similar to that observed in the cryo-electron tomography or patch clamp studies, it is also possible that oligomers could assemble from Aβ42 molecules that never left phospholipid membranes. In recent years, the proteolytic processing of APP substrate by γ-secretase has been revealed to be much more complex than previously imagined [42]. The initial cut is near the cytoplasmic side of the APP transmembrane domain, releasing the APP intracellular domain and forming a 48- or a 49-residue Aβ peptide. These long Aβs contain most of the APP transmembrane domain and are not secreted. Instead, they are further proteolyzed by the γ-secretase complex, trimming them from the C-terminus in increments of three or four amino acids. Thus, two general pathways to secreted Aβ peptides are Aβ49 → Aβ46 → Aβ43 → Aβ40 and Aβ48 → Aβ45 → Aβ42 → Aβ38. The predominant products are Aβ40 and Aβ42 and are largely secreted, but an unknown amount may remain in the membrane, particularly of the more hydrophobic Aβ42.

Hardy [43] has proposed that normal membrane clearance processes become less efficient with aging, allowing this membrane-bound Aβ to accumulate and toxicity to arise, possibly from oligomer pores (Figure 6). The accumulated Aβ would lead either directly or indirectly to toxic tau tangle formation, and microglia and proteins such as TREM2 would play a key role in resolving such damage through clearance of membrane-bound Aβ. Microglia are a very dynamic population of functionally different types of cells, ranging from a resting homeostatic population that constantly surveil the brain, to a responsive or activated population that, among other functions, results in phagocytosis of damaged neurons including those with membrane-bound Aβ aggregates [44,45]. TREM2 is a receptor on the surface of microglia that activates this population. Microglial activation is, at least to some extent, a protective event, and reduced activation results in a significantly enhanced risk of AD [46]. A TREM2-dependent protective response is triggered by deposition of amyloid seeds (small aggregates of Aβ), which drive the further aggregation and deposition of Aβ [47]. Biologically active preamyloid Aβ seeds, possibly oligomers or protofibrils, are present in vivo before Aβ aggregation and deposition become detectable with current methods. Monoclonal antibodies against several forms of preamyloid seeds were injected into APP transgenic mice before amyloid deposition became detectable, and one (aducanumab) led to significant reduction of Aβ deposition and downstream pathologies six months later [47]. Moreover, amyloid seeding is clearly enhanced in the absence of functional TREM2 [48]. Thus, genetic variability in microglial genes could contribute to the rate of accumulation of membrane-bound Aβ and to how quickly an individual progresses toward neuronal death.

## Figures and Tables

**Figure 1 molecules-27-08804-f001:**
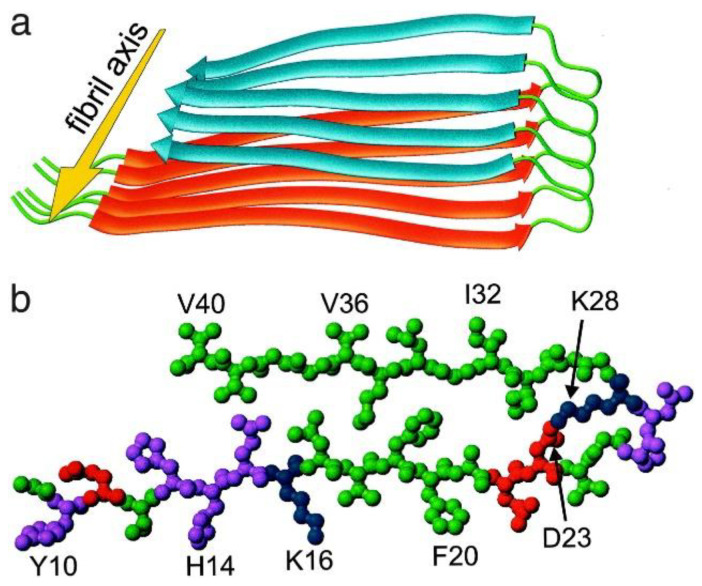
Structural model for Aβ40 fibrils, consistent with solid-state NMR constraints on the molecular conformation and intermolecular distances and incorporating the cross-β motif common to all amyloid fibrils [17]. Residues 1–8 are considered fully disordered and are omitted. (**a**) Schematic representation of a single molecular layer or cross-β unit. The yellow arrow indicates the direction of the long axis of the fibril, which coincides with the direction of intermolecular backbone hydrogen bonds. The cross-β unit is a double-layered structure, with in-register parallel β-sheets formed by residues 12–24 (orange ribbons) and 30–40 (blue ribbons). Mass-per-length measurements from electron microscopy indicate that fibrils are formed by two parallel cross-β units. (**b**) Central Aβ40 molecule viewed down the long axis of the fibril. Residues are color-coded according to their sidechains as hydrophobic (green), polar (magenta), positively charged (blue), or negatively charged red).

**Figure 2 molecules-27-08804-f002:**
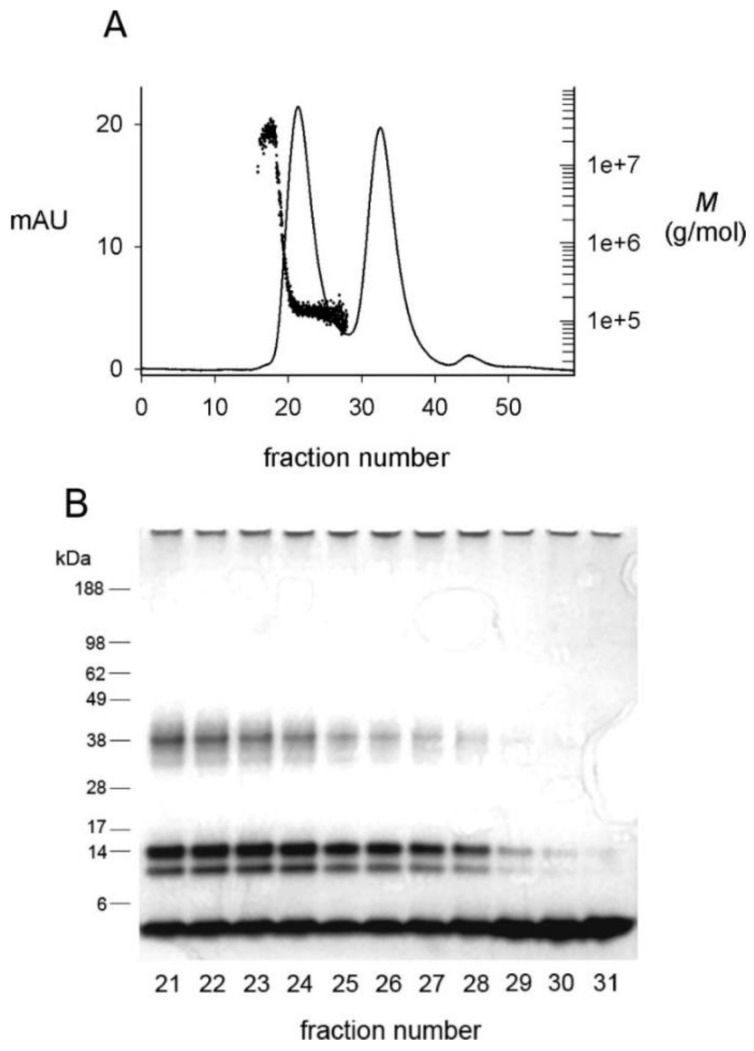
Fractionation of dialyzed Aβ(1–42) oligomers by size exclusion chromatography (SEC) [26]. SEC-purified Aβ(1–42) monomers (100 μM) were incubated in 4 mM SDS for 20 h and dialyzed against 10 mM Tris-HCl (pH 8.0) for 48 h at 25 °C to generate oligomers. (**A**) A 2-mL sample was applied to Superdex 75 equilibrated in 20 mM Tris-HCl (pH 8.0), and elution was monitored online by simultaneous recording of the absorbance at 280 nm (mAU, solid line) and the molecular weight (*M*) by multi-angle light scattering (dots). Peaks at fractions 21 and 33 corresponded to oligomers and monomers, respectively. Following a small number of larger aggregates near the void volume, the oligomer peak was characterized by a nearly constant *M* that averaged 150 ± 18 kDa. (**B**) Samples from the indicated fractions in panel A (50 pmol) were mixed with SDS gel loading buffer at 25 °C for PAGE analysis. The gel was stained with silver. Densitometry revealed a constant ratio of ~40 kDa, 14 kDa, 10 kDa, and 5 kDa (monomer) bands in fractions 21–28.

**Figure 3 molecules-27-08804-f003:**
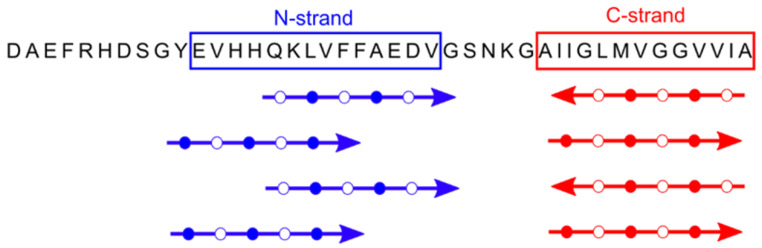
The primary and secondary structure of 150-kDa oligomers of Aβ42 (from data in [30]). Shown in blue and red are the β-strands detected by 2D NMR and the parallel or antiparallel organization of each β-strand. In addition, the registry shift within the N-strand β-sheet is likely to be three residues and alternate in direction (±3).

**Figure 4 molecules-27-08804-f004:**
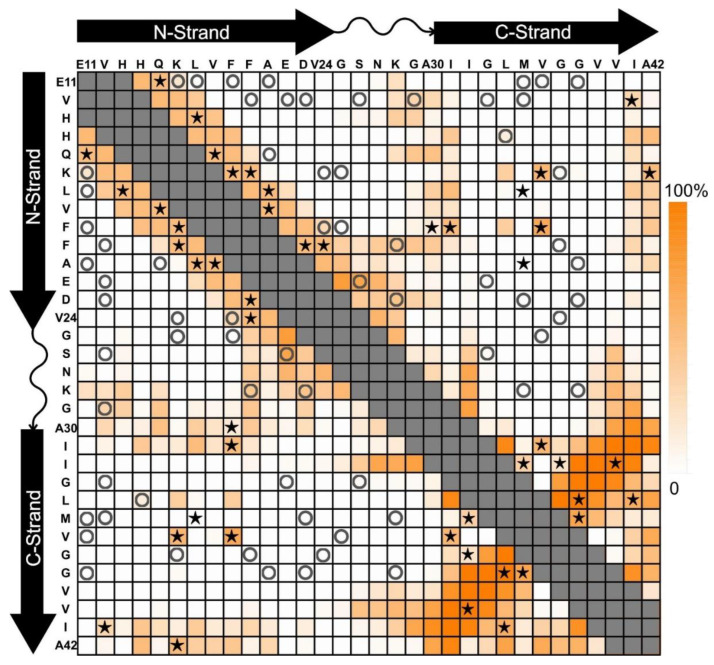
Chart of predicted and measured 2D DARR inter-residue contacts for the 150-kDa oligomer (see [30,34]). Symbols indicate when pairs of residues were isotopically labeled in the same sample and whether contacts were (stars) or were not (circles) observed. The color intensity scale (on right) indicates the model-predicted percentage that each residue pair would contribute to a 2D DARR-observable inter-residue contact in our model (see [34]). The conformation of the turn region (residues 24–29) and the interface between the β-sheets are not highly constrained. For pairs of residues corresponding to gray squares, intra-molecular ^13^C-^13^C couplings would make it difficult to interpret contacts in terms of β-strand alignment.

**Figure 5 molecules-27-08804-f005:**
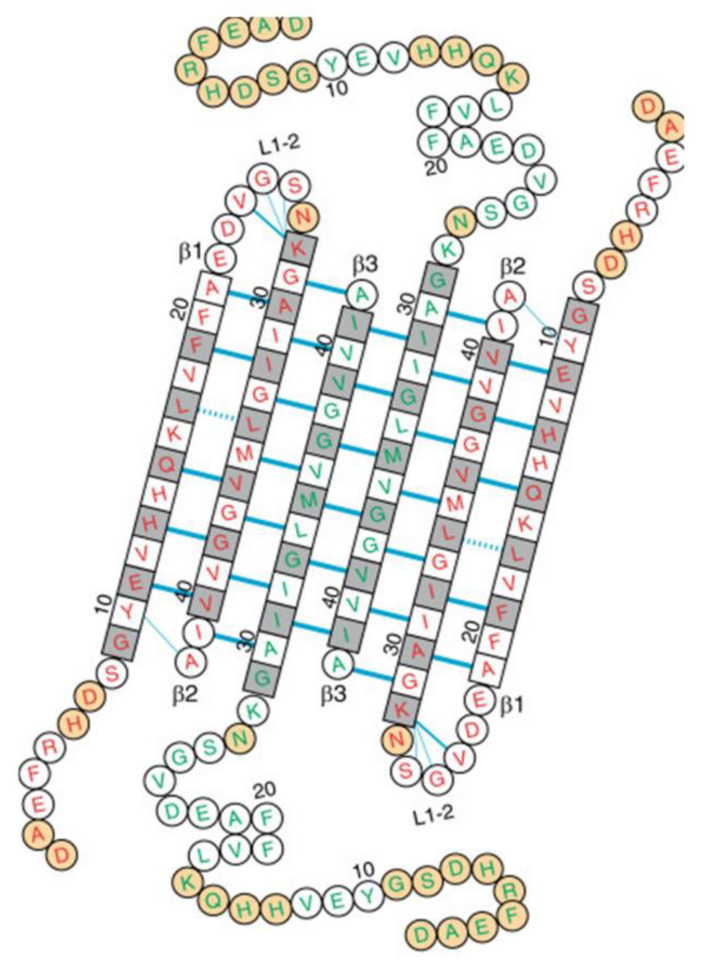
The amino acid sequence of the Aβ(1–42) tetramer arranged on the basis of the secondary and tertiary structure in the detergent DPC [37]. Amino acids in squares denote β-sheet secondary structure as identified by secondary chemical shifts; all other amino acids are in circles. Blue lines denote experimentally observed NOE contacts between two amide protons. The side chains of white and gray residues point towards distinct sides of the β-sheet plane. Orange circles correspond to residues that could not be assigned.

**Figure 6 molecules-27-08804-f006:**
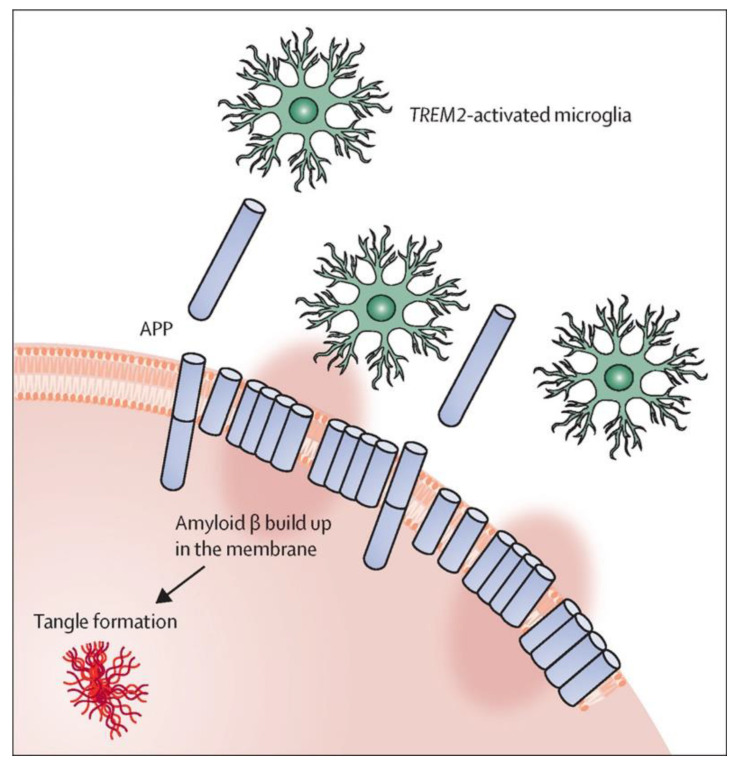
Amyloid-β neuronal damage and TREM2-mediated activation of microglia [43]. Amyloid-β deposition can disrupt the neuronal membrane and attract microglia, partly through TREM2 signaling, which, in healthy conditions, should repair or remove damaged membranes. When this process fails or is overwhelmed, tangle formation is instigated. The pink oval highlighting represents disruption of the membrane by the amyloid-β stubs.

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
