# Peer review of "Oligomer Formation by Amyloid-β42 in a Membrane-Mimicking Environment in Alzheimer’s Disease"

_molecules, 2022, doi:10.3390/molecules27248804_

Round 1
Reviewer 1 Report
The manuscript by Rosenberry et al., is focused on a specific aspect of molecular dissection of the amyloidogenic pathway in Alzheimer's disease. More than a review of the biological profile of beta-amyloid in AD, this is a recognition of the structural and biochemical state of the art of chemical characteristics of amyloid aggregates. The paragraphs do not offer a biological/biomedical view of aggregates in AD, reducing this work to a specific interest for biochemists. The role of microglia as players in the clearance of membrane-bound Ab is not sufficiently discussed and the conclusion is generic and does not mirror the review contents. Reconsider after major revision.
Author Response
Thank you for your review. Please refer to the attachment for details.

Reviewer 2 Report
In this manuscript, Rosenberry and colleagues describe recent progresses in amyloid-beta oligomer hypothesis. First, they show that the amyloid hypothesis is still a viable theory even though several high-profile clinical trials targeting amyloid fibrils or associated proteins show lack of efficacy and/or toxicity effects. From a diverse group of Ab aggregates, an apparent 150 kDa Ab42 oligomer could be stably produced in SDS-containing buffers. Second, authors discussed their efforts to characterize this Ab42 oligomer by solid-state 2D NMR and cryo-EM. Last, a link between the 150 kDa oligomer and a recently reported Ab42 4- or 8- mer formation in DPC was postulated. The cryo-EM 2D class average and structural model suggested a central pore that may explain the neurotoxicity induced by Ab42 oligomer in phospholipid membrane.
This review article provides useful information for the Ab oligomers and their involvement in AD. Several suggestions are provided.
1. The title is too broad when compared to the scope of the manuscript, which focuses on Ab oligomer.
2. The review is not balanced. More background in others’ work on Ab oligomers in aqueous or detergent environment is needed.
3. A figure showing the sequential cleavage pathway of APP may provide clear picture why Ab42 is more hydrophobic.
4. Figures 1, 2, 6 and 7 need copyright clearance.
5. Figures 4 and 5 are presumably from a pending manuscript by the authors. However, the reviewer did not find corresponding BioRxiv paper and could not provide assessment of the cryo-EM experiment.
Author Response
Thank you for your review. Please refer to the attachment for details. Please check the BioRxiv paper you mentioned in the fifth comment in non_published_material.

Round 2
Reviewer 1 Report
The authors responded to the reviewers' comments to improve the manuscript's readability. Accepted in present form.
Reviewer 2 Report
In this manuscript, Rosenberry and colleagues describe recent progresses in amyloid-beta oligomer hypothesis. First, they show that the amyloid hypothesis is still a viable theory even though several high-profile clinical trials targeting amyloid fibrils or associated proteins show lack of efficacy and/or toxicity effects. From a diverse group of Ab aggregates, an apparent 150 kDa Ab42 oligomer could be stably produced in SDS-containing buffers. Second, authors discussed their efforts to characterize this Ab42 oligomer by solid-state 2D NMR and cryo-EM. Last, a link between the 150 kDa oligomer and a recently reported Ab42 4- or 8- mer formation in DPC was postulated. The cryo-EM 2D class average and structural model suggested a central pore that may explain the neurotoxicity induced by Ab42 oligomer in phospholipid membrane.
Thank authors for the revision. This review article provides useful information for the Ab oligomers and their involvement in AD. To improve the manuscript further, here are a few minor suggestions.
1. A review article is supposed to be a summary of current “peer-reviewed” knowledge in the field. I would suggest that authors hold onto this manuscript till the acceptance of the BioRxiv paper by Gao et al. , which discussed the structural model for the 150 kDa Ab42 oligomer from a combinatory approach using NMR, Cryo-EM and molecular modeling.
2. Thank authors so much for sharing the Gao et al. paper. However, frankly, the Gao et al. paper can only be considered as a work-in-progress. In addition to missing figures, Methods are not disclosed properly for a critical review. In Results, Figure 2D only showed the top view of the proposed Ab42 oligomer. Is there any side view?